# RESONATOR-GATED RNNS

## ABSTRACT

Sequence learning tasks frequently involve data with repetitive and periodic temporal patterns. Detecting these patterns is essential for accurate predictions and informed decision-making in various domains. There is, however, still huge potential in augmenting sequence learning algorithms in this regard. In RNN-based sequence learning, gated RNNs, such as long short-term memory networks (LSTMs) and gated recurrent units (GRUs), are the de facto standard. While adept at capturing longer-term dependencies, gated RNNs still sometimes struggle with periodic data components, because their gating mechanism is designed to prioritize retaining transient relevant information. As a result, these networks often challenged by periodicity in the data. We present a novel memory unit that incorporates a simple resonator circuit. The resonator facilitates the recognition of periodic data patterns, focusing on data-specific time scales and respective frequencies. Moreover, it enables the forward propagation of information through resonating dynamics while stably channeling the gradient backwards. We show that our resonator-gated RNN (RG-RNN) accelerates the training convergence on multiple sequence classifications tasks. Moreover, it significantly outperforms vanilla LSTMs on three out of four benchmark tasks in terms of accuracy. We conclude that resonator-based gating offers a new inductive bias to gated RNNs, focusing learning on the detection and processing of periodic data patterns.

## 1 INTRODUCTION

Sequence learning involves various domains such as, speech recognition (Sutskever et al., 2014), biomedical signal analysis Nurmaini et al. (2021), or time series forecasting in climate science (Granata & Di Nunno, 2021). Models in these domains are mainly used for prediction, classification, generation, and comprehensive learning of and from time sequence information (Sutskever et al., 2014; Graves, 2013). Among the most popular techniques are recurrent neural networks (RNNs) which include long short-term memories (LSTMs) (Hochreiter & Schmidhuber, 1997) or gated recurrent units (GRUs) (Cho et al., 2014).

Data in this area of research often contains repetitive and periodic temporal patterns, e.g. seasonality in climate data. Regardless of the underlying RNN architectures, the ability to generate or differentiate these signals requires models to learn seasonal patterns, fundamental frequencies, and shifts therein. While gated RNNs are good at capturing longer-term dependencies over time, they often fail to recognize reoccurring or periodic patterns as they operate on fixed time scales or lack the necessary inductive biases (Neil et al., 2016; Liu et al., 2020). Therefore, targeting periodic signals can be a viable approach to boost performance in sequence learning tasks Neil et al. (2016); Huang et al. (2022).

These inductive biases can be found in a class of neurons called resonators. In contrast to integrators, resonators have the capability of responding to spike patterns that arise at specific frequencies rather than integrating over a short period of time. The same input experienced at different times can unfold both an inhibitory and excitatory effect. This effectively allows these neurons to naturally extract frequency patterns within the time domain and enables bridging of the temporal information gap if relevant information is sparsely scattered across time (Izhikevich, 2000; 2001).

While the value of resonator neurons in physiological neural networks has been discussed (Izhikevich, 2001; Tolmachev et al., 2018; AlKhamissi et al., 2021), the applicability in ANNs has not yet been investigated deeply. Examples of RNN extensions that facilitate the detection of periodic

signals indicate the demand for such capabilities (Neil et al., 2016; Huang et al., 2022). Similarly, recent advances in sequence models pushed the performance boundaries of RNNs (Gu et al., 2021; Smith et al., 2023; Orvieto et al., 2023).

We believe that the integration of resonators in ANNs and RNNs, in particular, can be of high interest for sequence learning, since their inductive bias has the potential to overcome the above mentioned shortcomings of gated-RNNs.

As a possible solution to fill this gap, we here propose the resonator-gated RNN (RG-RNN), a novel memory unit that incorporates a simple resonator circuit facilitating the recognition of periodic data patterns. We implement the RG-RNN by using an LSTM as a framework. This enables us to directly compare the models with almost identical hyper parameters and uses an established model with clear intuition as a carrier.

## 2 BACKGROUND

Resonator neurons respond favorably to incoming spike patterns matching their resonating frequency. This behavior can be modeled by a complex-valued state $z \in \mathbb{C}$ or equivalently by a two dimensional state vector $z \in \mathbb{R}^2$, where $z = [v, u]^T$, as opposed to the scalar state of an integrator neuron. The integration inside the resonator is coupled to the position of $z$ around its origin (Izhikevich, 2000; 2001).

In Izhikevich (2001) the resonate-and-fire neuron is defined by two differential state equations.

$$\frac{dv}{dt} = bv - \omega u \tag{1}$$

and

$$\frac{du}{dt} = \omega v + bu, \tag{2}$$

where $b \in \mathbb{R}_{\leq 0}$ and $\omega \in \mathbb{R}_{\geq 0}$ are parameters that represent the dampening and the resonating frequency of the resonator neuron, respectively. Input signals to the resonator can be handled in different ways. Izhikevich (2001) uses a simple summation in the real axis of the complex state. The resonators response can be measured by its excitation i.e. the magnitude of $z$.

Via Euler integration the differential state equations 1 and 2 can be transferred into discrete time with time step $t \in \mathbb{N}_0$ and step size $\delta \in \mathbb{R}_{\geq 0}$:

$$v^t = v^{t-1} + \delta(bv^{t-1} - \omega u^{t-1} + x^t) \tag{3}$$

and

$$u^t = u^{t-1} + \delta(\omega v^{t-1} + bu^{t-1}) \tag{4}$$

where $x^t$ is an input signal at time step $t$, that is added to $v^t$. The dependence on the previous state $v^{t-1}$ and $u^{t-1}$ acts as a local recurrence similar to internal cell state of an LSTM or the membrane potential regular spiking neurons.

Parameterizations of the resonate-and-fire neuron will express different behavior towards (periodic) signals including the capability of "skipping" the resonating mechanism entirely, thus being able to emulate integrators. Figure 1 shows the response of a resonator to an arbitrary input signal and the resonating frequency of the resonator through a Bode diagram. The resonating frequency generates the strongest response—strongest growth—in the resonator state. Because the integration of the differential equations 1 and 2 are directly dependent on $\delta$, it is a key parameter in this method.

RNNs utilize their recurrent connections to capture long-term dependencies in data. In practice, gated-RNNs like LSTMs (Hochreiter & Schmidhuber, 1997) or GRUs (Cho et al., 2014) are the de facto standard for gated RNNs. The incorporated gating allows information to be selectively passed through the control flow of the unit or forgotten if the need arises. The internal mechanisms are trained with back-propagation through time (BPTT) (Werbos, 1988), which is largely trivialized in modern automatic differentiation frameworks (Paszke et al., 2019).

The internal function of an LSTM is usually denoted with four equations for the gate activation and two equations for the state update. The gate activations are as follows

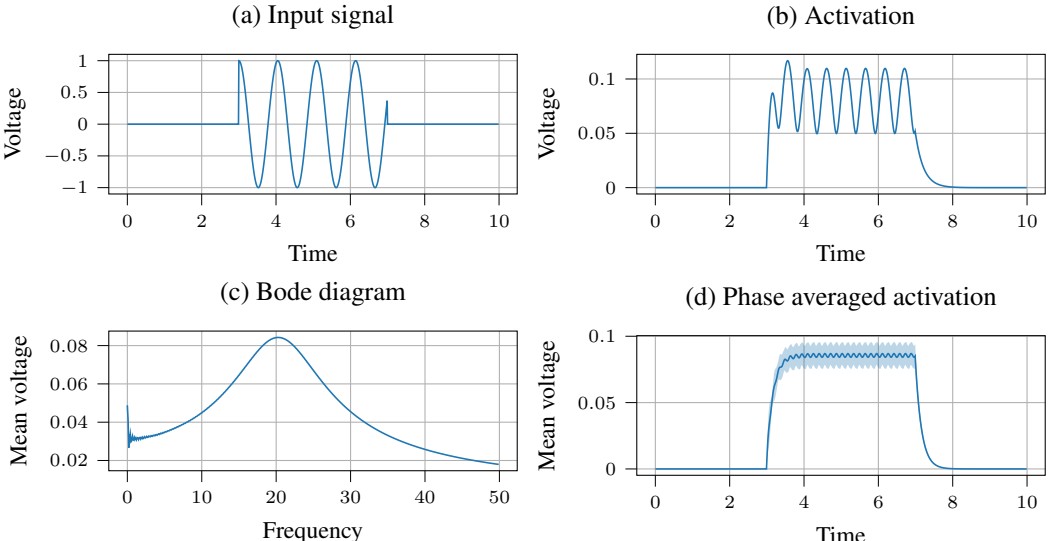

Figure 1: Responses of a resonator ($b = -7.808$, $\omega = 18.966$, $\delta = 0.01$) to input signals. **(a)** Input signal to the resonator. The periodic part is arbitrarily chosen. **(b)** Norm of the state, i.e. activation of the resonator. The resonator responds to the periodic input with a pattern similar to a square wave with oscillations. **(c)** Bode diagram of the resonator over frequencies in $f \in [0, 50]Hz$. It shows the averaged response to an input of similar structure as (a) with frequency $f$. The peak indicates the resonating, i.e. most favourable, frequency. **(d)** Averaged response of the resonator at its resonating frequency when the phase of the input is shifted randomly between $0$ and $2\pi$ for 200 samples. The small standard deviation (shaded area) demonstrates the invariance to phase shifts at the resonating frequency.

$$g^t = \tanh\left(x^t W_{g,i} + b_{g,i} + h^{t-1} W_{g,h}\right) \tag{5}$$

$$i^t = \sigma\left(x^t W_{i,i} + b_{i,i} + h^{t-1} W_{i,h}\right) \tag{6}$$

$$f^t = \sigma\left(x^t W_{f,i} + b_{f,i} + h^{t-1} W_{f,h}\right) \tag{7}$$

$$o^t = \sigma\left(x^t W_{o,i} + b_{o,i} + h^{t-1} W_{o,h}\right) \tag{8}$$

where $g^t$, $i^t$, $f^t$, and $o^t$ are the cell input, input gate, forget gate, and output gate, respectively. The current time step is denoted by $t$. The input is given by $x^t$ and the last hidden state by $h^{t-1}$. The parameters are contained in matrices $W_{\text{gate,connection}}$. To calculate the activation and hidden state of the LSTM the following equations are used

$$c^t = f^t \odot c^{t-1} + i^t \odot g^t \tag{9}$$

$$h^t = o^t \odot \tanh(c^t) \tag{10}$$

Here $c^t$ denotes the internal activation of the LSTM. In Equations 6-8 $\sigma$ refers to the sigmoid activation and $\tanh$ to the hyperbolic tangent activation functions, the output of the LSTM is calculated by iteratively processing each input step in the input sequence and evaluating the hidden state. The last hidden state or the full hidden state sequence can then be used as output. In multi-layer LSTMs the input of the next recurrent layer is the hidden state of the previous layer at the current time step.

## 3 RESONATOR-GATED RNN

Our definition and implementation of the resonator-gated RNN (RG-RNN) combines the properties and inductive biases of RNNs and resonator neurons by extending an LSTM with a resonator-gating unit. The resonator is derived and implemented based on Equations 3 and 4. Its state is calculated

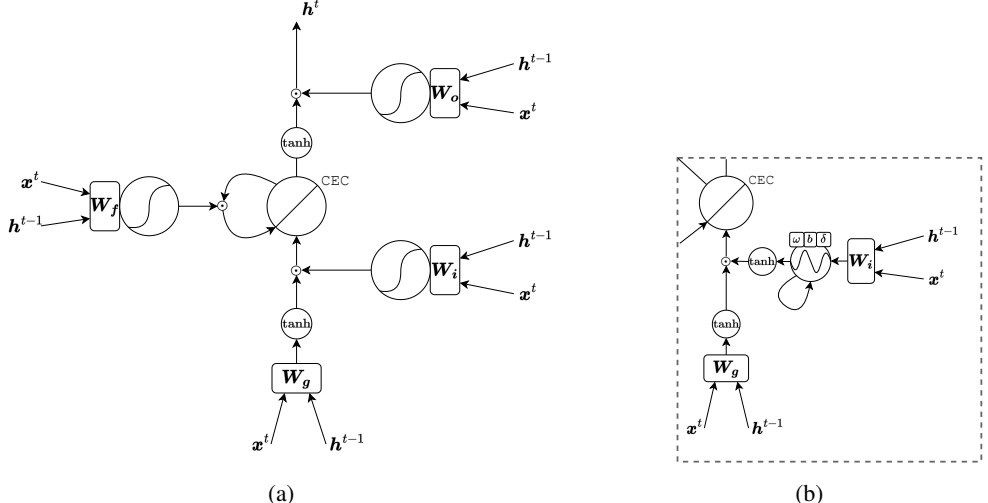

(a)                                                  (b)

Figure 2: **(a)** Schematic of the LSTM. It has four input paths for passing $\boldsymbol{x}^t$ and $\boldsymbol{h}^{t-1}$. $\mathtt{tanh}$ transformations are indicated with $\mathtt{tanh}$ and sigmoid activations with a stylized "S" in the nodes. Small nodes labeled $\odot$ depict element wise multiplication (Hadamard product). During the computation of step $t$ the CEC out-edge leading towards the forget gate holds the activation from the previous step $t-1$. **(b)** Extension of the LSTM with a resonator. Notation and schematic are similar to Figure 2a. The resonator neuron is placed after the affine transformation of the input gate and before the activation. Because the resonator output is $>= 0$, the activation is replaced by a $\mathtt{tanh}$ activation to cover the range $[0,1]$

by

$$\boldsymbol{v}^t = \boldsymbol{v}^{t-1} + \boldsymbol{\delta}(\boldsymbol{b}\boldsymbol{v}^{t-1} - \boldsymbol{\omega}\boldsymbol{u}^{t-1} + \boldsymbol{x}_r^t) \tag{11}$$

$$\boldsymbol{u}^t = \boldsymbol{u}^{t-1} + \boldsymbol{\delta}(\boldsymbol{\omega}\boldsymbol{v}^{t-1} + \boldsymbol{b}\boldsymbol{u}^{t-1}) \tag{12}$$

$$\boldsymbol{z}^t = \begin{bmatrix} \boldsymbol{v}^t & \boldsymbol{u}^t \end{bmatrix} \tag{13}$$

and its activation by

$$\boldsymbol{r}^t = |\boldsymbol{z}^t| - \|\boldsymbol{\delta}\|_2 \tag{14}$$

with $|\boldsymbol{z}^t|$ being the magnitude of the two dimensional state and not the full vector.

The parameters $\boldsymbol{b}$, $\boldsymbol{\omega}$, and $\boldsymbol{\delta}$ are trainable and initialized randomly in the ranges $(0,1)$ for $\boldsymbol{b}$ and $\boldsymbol{\omega}$, and $(0.01, 0.1)$ for $\boldsymbol{\delta}$. To guarantee that the parameters adhere to the numerical ranges described in Section 2 they need to be transformed accordingly. This was achieved by taking the absolute value of the real underlying parameters $\hat{\boldsymbol{b}}$, $\hat{\boldsymbol{\omega}}$, and $\hat{\boldsymbol{\delta}}$ and further negating it in the case of $\boldsymbol{b}$.

Because the resonator unit is derived from the Euler integrated differential equations, its stability greatly depends on the parametrization of $\boldsymbol{\delta}$. To aid the stability of the resonator, the magnitude of the resonator state is regularized by the magnitude of $\boldsymbol{\delta}$ itself. This reduces the potential for $\boldsymbol{\delta}$ to grow during training, destabilizing the resonator. In initial tests, which omitted the regularization term, the state of the resonator was able to grow into regimes where the numerical range of floating point numbers was exhausted and even simple random input signals would drive the resonator into self-oscillation with $\boldsymbol{\delta} \geq 0.8$.

$$\boldsymbol{b} = -|\hat{\boldsymbol{b}}|, \ \boldsymbol{\omega} = |\hat{\boldsymbol{\omega}}|, \ \boldsymbol{\delta} = |\hat{\boldsymbol{\delta}}| \tag{15}$$

The resonator unit is implemented as a differentiable module using $\mathtt{PyTorch}$ (Paszke et al., 2019). It can be inserted into the gating of an LSTM, where we can keep track of the resonator state and execute it as part of the forward pass of the LSTM. The control flow of the implemented unit is depicted in Figure 2b.

The resulting RG-RNN modifies LSTM Equation 6 as

$$\boldsymbol{i}^t = \tanh\left(\boldsymbol{r}^t\right) \tag{16}$$

with resonator input

$$\boldsymbol{x}_r^t = \boldsymbol{x}^t \boldsymbol{W}_{i,i} + \boldsymbol{b}_{i,i} + \boldsymbol{h}^{t-1} \boldsymbol{W}_{i,h} \tag{17}$$

The resonator sits after the affine transformation and before the non-linearity of the LSTM gate (Figure 2b). The activation of the gate was converted to a $\tanh$ activation, because the output $\boldsymbol{r}$ is always positive. With a sigmoid activation the range would be restricted to $[0.5, 1]$ while providing a gradient even when $\boldsymbol{z} = 0$.

Note that the integration of the resonator as a building block is of course not limited to LSTMs. It could, for instance, also be integrated into the GRU model replacing the update gate. Moreover, it is thinkable to implement the resonator as a standalone layer, which could be flexibly arranged within RNN architectures.

## 4 EXPERIMENTS

The RG-RNN was evaluated against an LSTM baseline. For each experiment and model type five models were trained from scratch and evaluated using the mean loss and mean accuracy of the five runs. The training was based on a per-time step evaluation i.e. the models calculated a prediction after each presented time step. The labels were either uniform across all time steps or a segmentation, in the case of the QT Database (QTDB[1]). We found that using a single-label approach, where the training is only based on the last time step, did not deviate greatly from the results of the per-time step training. We used the cross-entropy loss as the training criterion and RMSprop (Hinton, 2018) as the optimizer during training. Between 1 and 2 recurrent layers followed by a single linear layer were used. The input to the linear layer was the hidden state sequence of the recurrent layers which was input per time step. The full list of model hyper parameters is provided in Table A1.

### 4.1 DATASETS

To evaluate the implementation of the RG-RNN we trained LSTMs and RG-RNNs on three sequence datasets chosen from the benchmarks presented in Yin et al. (2021) and a dataset of time-series sensor data. The datasets were sequential MNIST (SMNIST), including permuted sequential MNIST (PMNIST) (Le et al., 2015), Speech Commands V2 (Warden, 2018), QTDB (Goldberger et al., 2000), and a multivariate gait analysis (MGA) dataset from Helwig & Hsiao-Wecksler (2022).

#### 4.1.1 SMNIST AND PMNIST

MNIST is a dataset consisting of hand written digits collected by the US postal service. Each sample is a 28 by 28 pixel gray scale image. The associated label of each sample represents the digit.

The SMNIST variation of MNIST is a popular benchmark for sequence models. In SMNIST the $28{\times}28$ pixel images are converted into sequences of length $T = 28 \cdot 28 = 784$ where the rows (or columns) are concatenated one after another.

PMNIST raises the complexity of the problem by permuting the 784 steps of the sequence in a random fashion. While the permutations are chosen randomly it is necessary to fix the permutation for each run since learning the problem would otherwise be impossible.

The periodicity in SMNIST is expressed through the regular arrangement of signals in the samples. The data is condensed in the center of each image which leads to patterns when concatenating the rows into sequences. In PMNIST this periodicity is deliberately disrupted through the random permutation. This removes the original causal relationship between signal and placement in the sequence.

While the loss is calculated based on each time step the accuracy is calculated using only the last prediction of the model.

---

[1]QT referres to the QT waveform interval in the ECG

### 4.1.2 SPEECH COMMANDS V2

The speech commands V2 dataset contains one second audio recordings. The recordings contain word utterances recorded across a wide range of devices. Each sample is labeled with on of 21 classes representing the uttered word. The dataset includes utterances which do not correspond to a class label but are instead placed into the *unknown* class. The classes are: *unknown*, *yes*, *no*, *up*, *down*, *left*, *right*, *on*, *off*, *stop*, *go*, *zero*, *one*, *two*, *three*, *four*, *five*, *six*, *seven*, *eight*, *nine*.

The task of the models is to predict the class of the utterance. As a pre-processing step the raw 16000Hz audio signals are converted into MFCC frames. The length of the sequences would otherwise prevent the models from learning anything meaningful in a feasible amount of time (Sutskever, 2013). The conversion was performed using `torchaudio`. The training setup was roughly based on Bernardo et al. (2020). We modified the parameters of the MFCC transformation to produce sequences of length 801 because we estimated that this would be most beneficial for the learning of meaningful temporal dynamics. The complete set of parameters for the MFCC feature extraction can be taken from Table A3.

The conversion into MFCC frames transforms the problem such that the underlying frequencies are not immediately apparent. This poses an additional challenge for the RG-RNN. In its un-processed form Speech Commands V2 presents itself as an interesting study of inherently periodic data that is unfortunately computationally infeasible. The conversion into MFCC spectra transforms the problem such that the underlying frequencies are not immediately apparent. To leverage the resonator, repeating patterns in frequency bands have to be present and extracted by the RG-RNN.

While the loss is calculated based on each time step the accuracy is calculated using only the last prediction of the model.

### 4.1.3 QTDB

The last dataset from Yin et al. (2021) is generated from QTDB Goldberger et al. (2000). QTDB holds ECG records collected from different resources for a total of 105 patients. The ECG signals are two channel 15 minute recordings, recorded at or converted to a 250Hz sampling rate and annotated by two experts for a selection of the beats in each recording. In addition to the expert annotations machine annotations were calculated for each sample. Similar to Yin et al. (2021) we use sequences of length 1300. We omit the termination signal that is included in Yin et al. (2021). To ensure the best possible quality of the training data we only considered the expert annotated samples of expert one in our experiment. Records from *Sudden Death* were excluded in the training similar to Nurmaini et al. (2021) since they show vast differences to all other signals. Additionally, record *sel232* was excluded due to the use of the undocumented label *A*.

The remaining signals were cut into 1300 long sequences starting from the first annotation, until the end of the sequence. The last segment was omitted if it was shorter than 1300 steps, additionally all segments that only contained 0-label steps, i.e. unannotated baseline signals, were also omitted to reduce class imbalance. The class weights to compensate against the remaining class imbalance can be found in Table A2. Lastly, there were cases were the annotations had large gaps, which can be observed in sample *sel104*. This lead to challenges to automate the pre-processing and selection of samples, because the signal which contained the proper wave forms was not fully annotated. This would lead to problems in training. For these cases the samples were manually sighted and selected to exclude such cases. The total number of samples in the resulting dataset was 379 which was further split into a train and test set with a 80/20 split.

The task was to predict the current label that represents the phase of the ECG signal, i.e. *p*, *N*, *t*, or *u* representing the corresponding complex (Ashley & Niebauer, 2004).

ECGs hold the possibility to learn both the periodic components of the signal itself and the patterns of the recurring complexes across the sequence.

Both the loss and the accuracy are calculated per-time step as opposed to the other datasets.

Table 1: Average test results of all experiments after the final training epoch. For an exhaustive list of all collected metrics see Table A4

| Benchmark | LSTM | RG-RNN (ours) |
|---|---|---|
| SMNIST | 98.44% | **98.51%** |
| P-SMNIST | 89.49% | **93.55%** |
| QTDB | 72.86% | **84.46%** |
| Speech Commands V2 | **92.30%** | 91.79% |
| MGA | **96.83%** | 96.62% |

## 4.2 MGA

For an additional diversification of the experiments the MGA dataset was analysed. It consists of the measured joint angles collected from ten subjects during walking on a treadmill. Each subject performed ten repetitions i.e. step cycles and was measured under three conditions. (1) unbraced, (2) with a knee brace, and (3) with an ankle brace. The samples were constructed by concatenating the angles for a subject under a repetition into a vector $x \in \mathbb{R}^{6,101}$ where 6 is the number of joint angles (three per leg) and 101 is the number of time steps in a repetition. The task is to classify the condition of the subject.

## 4.3 RESULTS

The training results on all experiments follow a similar pattern. We observe a fast decline in the loss and rise in the accuracy, immediately superseding the LSTM in both measures. This can be observed in all experiments in Figure 3 and 7 during the first epochs. Additionally, the training stability is vastly improved which is indicated by the small error bands around the training trajectory. The difference compared to the LSTM is most clearly visible in the SMNIST and PMNIST experiment in Figures 3, 8, and 9. The overall performance also seems to benefit from the addition of the resonator. The model performance is recorded in Table 1 and Table A4 for all measures. While it is not possible for the RG-RNN to outperform the LSTM in the Speech Commands V2 experiment the other two experiments show improvements in both accuracy and loss. It is to note that the difference in both the SMNIST and the Speech Commands V2 experiment was small but occurred consistently over multiple evaluations. The experiment on MGA in Figure 7 shows a similar convergence profile as the previously discussed experiments. The results suggest that the problem itself is easily handled by both LSTM and RG-RNN. More detailed evaluation and convergence curves can be taken from Figures 8, 9, 10, and 11 in Appendix A.4.

In addition to these standard benchmarks we performed additional supplemental experiments on SMNIST. We studied the robustness against recurrent sparsity which gives indications on the self-sufficiency of the internal state by restricting access to memory. The curves for the RG-RNN in Figure 5 largely retain a similar pattern as in the SMNIST experiment across all sparsity rates. A slight impact to convergence and performance can be observed starting at 75% sparsity with an even more pronounced impact when no recurrences are retained, essentially blocking the access to memory. The LSTMs performance was heavily influenced by all sparsity rates, as evident from Figure 6. First slightly decreasing the convergence and increasing the variance between runs at 25% and 50% sparsity. At 75% sparsity the convergence is suprisingly improved with only a small impact in accuracy. Lastly, removing all recurrent connections has the greatest impact on the LSTM severly reducing its accuracy. The performance of the RG-RNN and the LSTM can be taken from Tables A6 and A7, respectively. Additional information on this experiment is provided in Appendix A.3.

## 5 DISCUSSION

The experiments on SMNIST have shown a great response of the resonator to highly regular inputs. The structure of the samples in MNIST places the signals into predictable intervals corresponding to the center of the image. This can be exploited by the resonator through its ability to anticipate signals, leading in principle to a fast training result and fast convergence. An analytical analysis presented in Appendix A.2 further suggests that the states gradient flow is best when following the

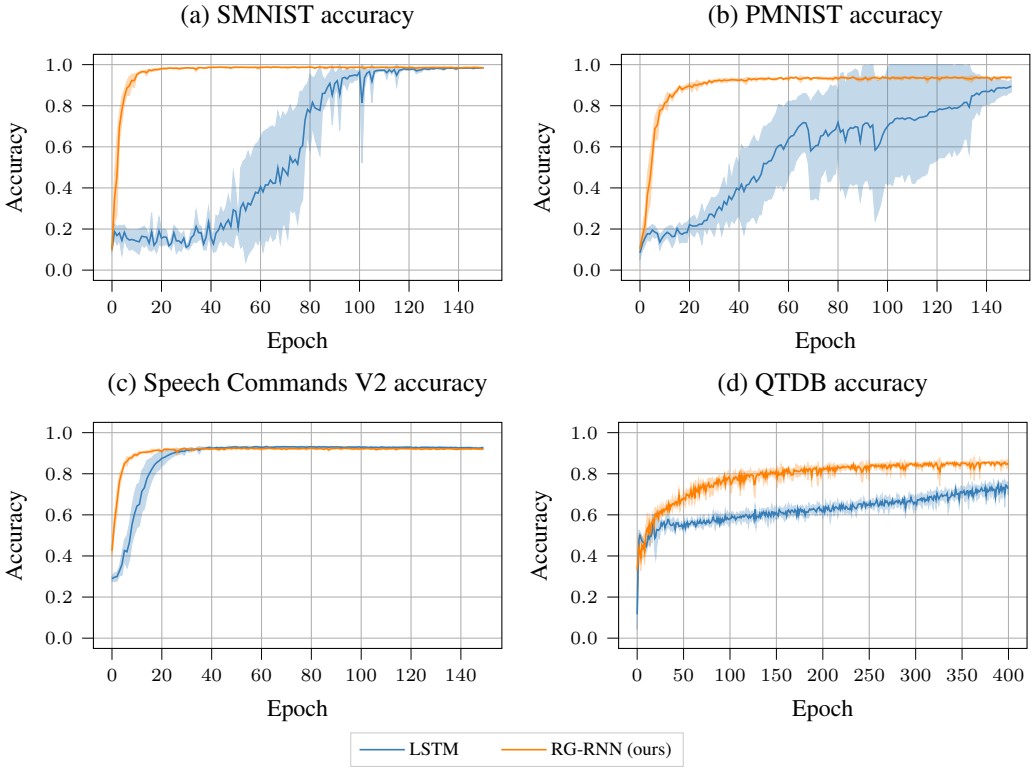

Figure 3: Averaged test accuracy of the main experiments as described in Section 4. The orange curve depicts the RG-RNN and the blue curve the LSTM. The shaded area is the standard deviation between recorded runs. The curves show similar patterns across all four experiments. There is a sharp rise in performance during the first few epochs with an early convergence for the RG-RNN, compared to the LSTM. Additionally, the variance between each run is almost non-existing, while the LSTM shows large deviations in curves **(a)**, **(b)**, and **(c)**.

parametrized frequency, because the resonator poses as an additional CEC that rotates the gradient based on $\delta$ and $\omega$.

We were pleasantly surprised to see the RG-RNNs convergence and performance on PMNIST to follow a similar trend as in SMNIST. While the regular structure of the samples is disrupted, the RG-RNN is still able to quickly converge to a level that is almost on-par with current SotA-RNNs on PMNIST Wisdom et al. (2016); Cooijmans et al. (2016). Additionally, we suspect that the use of a the sequence-labeled training regime (Section 4) leads to faster propagation of information compared to a conventional single-label training. Nevertheless, because the recorded performance of the LSTM is comparable to the values recorded in literature Arjovsky et al. (2015) we consider our approach as a valid alternative.

The improved convergence and performance on QTDB demonstrates the practical applicability of resonators and the RG-RNN on real-world data. The added inductive bias appears to facilitate the detection of key frequencies in the ECG samples. At the same time, the convergence behavior of the RG-RNN on the speech commands dataset reveals a limited advantage here. This is due to the fact that the preprocessing already extracts frequency components in form of short-term spectra. Repetitive patterns have to be extracted from within these spectrograms and the time signal directly. This suggests that, although computationally demanding, future experiments should also be performed on raw audio data to correctly access the impact of resonators on speech detection.

These results were achieved with only a small addition of trainable parameters as can be taken from Table A5 and compared to a model with an increased number of parameters in Figure 4. This further suggests that the inductive bias encoded by the resonator-gating is responsible for the

improved performance rather that just an increase in parameters as is often suspected (Frankle & Carbin, 2018).

In addition to the capability to respond to specific frequency ranges within the data, another advantage of RG-RNNs is that the resonator effectively pursues an anticipatory forward propagation of information through time via its local recurrence. This shields the information from outer disturbances and noise. At the same time, the resonator provides a "gradient highway" in backward direction similar to CEC-forget gate dynamics in LSTMs but with a constant period, which seems to stabilize the overall training behavior further.

Over the last decade advances in sequence learning gave rise to new models that differ from the classic gated RNNs, namely Transformers (Vaswani et al., 2017), temporal convolutional networks (TCNs) (van den Oord et al., 2016) and more recently state space model (SSM) such as S4 (Gu et al., 2021), S5 (Smith et al., 2023), and similar. While transformers are powerful due to the use of the attention mechanism and their ability to process sequences in parallel, the computation can be costly as they scale in the order $\mathcal{O}(n^2)$ with the input size (Vaswani et al., 2017; Gu et al., 2021; Orvieto et al., 2023). TCNs utilize dilated convolutions and deep networks to build powerful embeddings (van den Oord et al., 2016). In S4 and S5 the model is derived from SSMs and leverages HiPPO theory and the diagonalization of complex matrices to achieve high performance on long range dependency tasks without the use of attention (Gu et al., 2021). Interestingly, slight parallels can be drawn from the recurrent matrix in S4 to the resonator when rewriting the state in Equation 13 with complex parameters. If and which relation there is to SSMs can be an interesting approach to explore in future works that focus on the theoretical embedding of resonators. Lastly, the recently published linear recurrent unit (LRU) (Orvieto et al., 2023) aims to solve known issues in RNNs by applying insights from S4-based models to linear RNNs. LRUs bring a number of enhancements into the vanilla RNN framework that increase the parallelization and especially performance on long-range dependencies. This promising advancement in sequence learning and in particular RNN literature can pose as an opportunity for research to focus more on applications that employ RNNs and may offer an option to integrate resonators into LRUs. Future work could extend on the possibility of integrating the RG-RNNs inductive bias into LRUs as well as conduct comprehensive performance studies of the RG-RNN against current baselines.

## 6 CONCLUSION

We have introduced the resonator as a means to add to the inductive bias of RNNs by enabling the detection and response to periodic signals in sequence data. The implemented RG-RNN shows improved convergence on all tested benchmarks and better performance in three out of four experiments. Even when obvious patterns in the data are disrupted through randomization or the ablation of recurrent connections the RG-RNN can still learn and retain relevant information. All achieved with only a small change in the number of trainable parameters.

We conclude that resonator-based gating offers a strong, new inductive bias for gated RNNs that is applicable to a variety of problems, but excels at detecting patterns and periodicity in the input sequence. This can be attributed to the addition of an additional internal CEC which enables the interplay of rotation and decay and opens the possibility to tune into periodicity in the data.

The resonators introduce an anticipatory forward propagation, effectively bridging information gaps across time, which we propose to be beneficial for the overall stability of the training procedure. In addition, the internal dynamics of the resonator appear to allow the RG-RNN to learn more effectively without the utilization of a memory.

As future work we intend to explore possible synergies of RG-RNNs embedded within larger-scale recurrent architectures, balancing resonating and non-resonating components, as well as the use of smaller resonator units without the need to be embedded in LSTMs. Additionally, a rigorous comparison of performance of RG-RNNs with current S4, LRUs, or Transformers and the integration of resonators into these models can show the standing of resonator-gated and resonator-enhanced models in practice.

REPRODUCIBILITY

We actively try to contribute to make research more reproducible. In this effort we want to disclose all information necessary to reproduce the work in this paper. Sections 2 and 3 cover the necessary basis to implement the introduced method. Section 4 as well as Tables A2, A3, and A1 include the information needed to reproduce the performed experiments and benchmarks. Lastly, the Figures in Section 4 and in the Appendix provide information about the convergence of the implemented models. In addition we plan to release the code used for all experiments and implementations publicly to GitHub.

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

# A  APPENDIX

## A.1  HYPER PARAMETERS

The following Tables A1, A2, and A3 show the hyper parameters used to achieve the results presented in Section 4. Table A4 lists the full list of test statistics gathered during training and Table A5 lists the number of trainable parameters for the trained LSTM and RG-RNN as well as the number of trainable parameters for an LSTM with an additional hidden unit per-layer. An experiment on SMNIST which considers this is presented in Figure 4.

Table A1: Model hyper parameters for all performed experiments. The hyper parameters were the same for LSTM and RG-RNN which lead to a small increase in the number of trainable parameters for the RG-RNN. The number of trainable parameters is listed in Table A5.

| Hyper Parameter | S/P-MNIST | Speech Commands V2 | QTDB | MGA |
|---|---|---|---|---|
| Layers | 1 LSTM, 1 Linear | 1 LSTM, 1 Linear | 2 LSTM, 1 Linear | 1 LSTM, 1 Linear |
| Hidden units | 128 | 128 | 512 | 32 |
| Batch size | 64 | 64 | 128 | 64 |
| Learning rate | 0.0005 | 0.0005 | 0.0001 | 0.0005 |
| Epochs | 150 | 150 | 400 | 200 |

Table A2: Class weights calculated on the training set during the QTDB experiment. Weights are calculated by dividing the number of labels of any class against the number of labels of the largest class (0-label).

| Class | Weight |
|---|---|
| 0 | 1.0000 |
| p | 3.4879 |
| N | 3.7712 |
| t | 1.3838 |
| u | 7.5885 |

Table A3: Set parameters for MFCC pre-processing of the Speech Commands V2 experiment. The features were extracted using the `torchaudio` library version 2.0.2. Unreferenced parameters were kept at default values.

| Parameter | Value |
|---|---|
| # FFT features | 160 |
| # MFCC features | 20 |
| # Mel filter banks | 20 |
| Hop size | 1.25ms/20 steps |
| Window length | 10ms/160 steps |

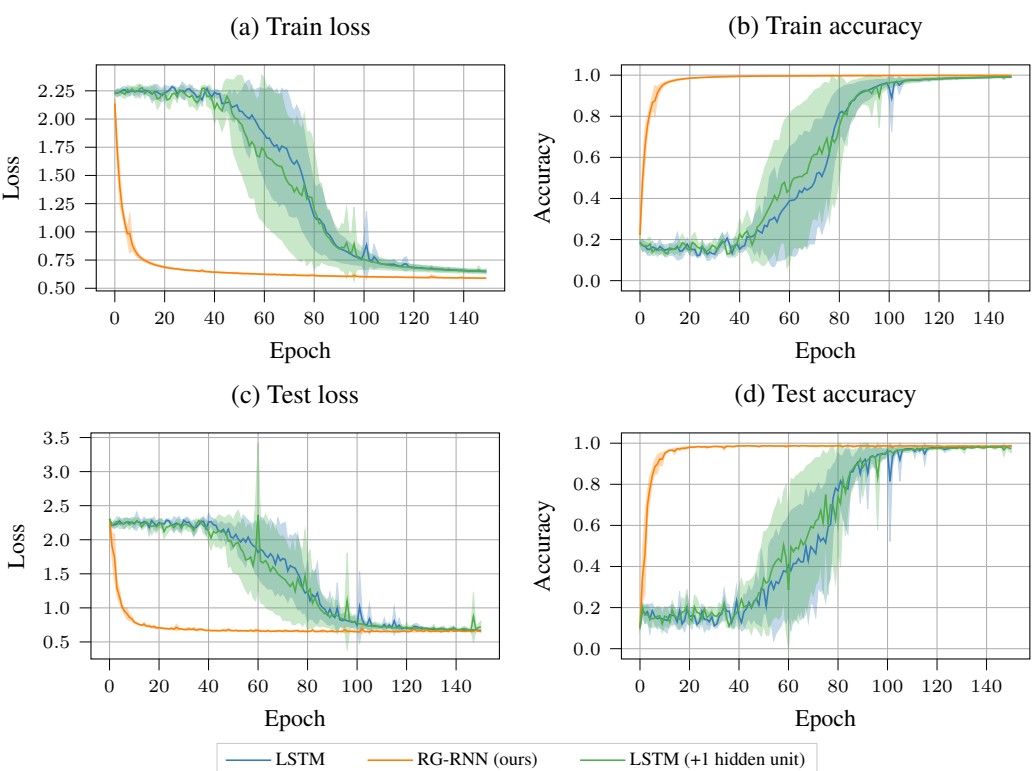

Figure 4: **SMNIST** experiment including an LSTM with additional hidden unit. Shown are averaged measures and standard deviation of the LSTM (blue), the RG-RNN (orange), and the LSTM with the additional unit (green) for (a) train loss, (b) train accuracy, (c) test loss, and (d) test accuracy.

Table A4: Average results of all experiments after the final training epoch. An explicit validation set was only given in the Speech Commands V2 experiment. Please note, that the test set of the SMNIST and the QTDB data sets was used in the same fashion as one would usually use a validation set.

| | SMNIST | | PMNIST | | QTDB | | Speech Commands V2 | | MGA | |
|---|---|---|---|---|---|---|---|---|---|---|
| Metric | LSTM | RG-RNN | LSTM | RG-RNN | LSTM | RG-RNN | LSTM | RG-RNN | LSTM | RG-RNN |
| Train loss | 0.6491 | **0.5898** | 0.3086 | **0.1625** | 0.4435 | **0.1696** | **0.9466** | 0.9678 | 0.1298 | **0.1013** |
| Train accuracy | 99.14% | **99.85%** | 93.32% | **99.09%** | 75.69% | **90.74%** | **98.24%** | 97.59% | 95.54% | **96.67%** |
| Test loss | 0.6681 | **0.6574** | 0.4484 | **0.3926** | 0.5370 | **0.4319** | **1.1918** | 1.2006 | **0.1155** | 0.1244 |
| Test accuracy | 98.44% | **98.51%** | 89.49% | **93.55%** | 72.86% | **84.46%** | **92.30%** | 91.79% | **96.83%** | 96.62% |
| Validation loss | - | - | - | - | - | - | **1.1887** | 1.1919 | | |
| Validation Accuracy | - | - | - | - | - | - | **92.67%** | 91.97% | | |

Table A5: Number of trainable parameters by experiment for the LSTM, RG-RNN and an LSTM with an additional hidden unit.

| Model | P/S-MNIST | SpeechCommandsV2 | QTDB | MGA |
|---|---|---|---|---|
| LSTM | 68362 | 79509 | 3160581 | 5219 |
| RG-RNN | 68746 | 79893 | 3163653 | 5315 |
| LSTM (+1 Hidden Unit) | 69412 | 80646 | 3172910 | 5514 |

## A.2 ON THE RESONATOR'S INTERNAL GRADIENT FLOW

One explanation of the cleaner and steeper convergence of the RG-RNN compared to the convergence of LSTMs, may be that the resonator itself establishes a similar but more advantageous structure of a constant error carousel (CEC).

Analogously to the input of the resonator being passed or blocked depending on the current resonator state $z^t$, the gradient that flows through the resonator underlies the same principle. However, the more important question of what happens to the gradient within the resonating circuit remains.

For simplicity consider the following equations for just a single resonator cell:

$$v^t = v^{t-1} + \delta(bv^{t-1} - \omega u^{t-1} + x_r^t) \tag{18}$$

$$u^t = u^{t-1} + \delta(\omega v^{t-1} + bu^{t-1}) \tag{19}$$

$$\mathbf{z}^t = \begin{bmatrix} v^t & u^t \end{bmatrix} \tag{20}$$

To uncover the local state change dependency, we derive the resonator state $z^t$ with respect to the previous state. $u$ and $v$ additionally influence themselves through the explicit recurrent connections in the network, that is, the output of the resonator is part of input to the resonator in the next time step. For simplicity we ignore these outer recurrencies.

We get:

$$\frac{\partial \mathbf{z}^t}{\partial \mathbf{z}^{t-1}} = \begin{bmatrix} \frac{\partial v^t}{\partial v^{t-1}} & \frac{\partial v^t}{\partial u^{t-1}} \\ \frac{\partial u^t}{\partial v^{t-1}} & \frac{\partial u^t}{\partial u^{t-1}} \end{bmatrix} \tag{21}$$

where

$$\frac{\partial v^t}{\partial v^{t-1}} = \frac{\partial}{\partial v^{t-1}} \left( v^{t-1} + \delta(bv^{t-1} - \omega u^{t-1} + x_r^t) \right)$$
$$= 1 + \delta b \tag{22}$$

$$\frac{\partial v^t}{\partial u^{t-1}} = \frac{\partial}{\partial u^{t-1}} \left( v^{t-1} + \delta(bv^{t-1} - \omega u^{t-1} + x_r^t) \right)$$
$$= -\delta \omega \tag{23}$$

$$\frac{\partial u^t}{\partial v^{t-1}} = \frac{\partial}{\partial v^{t-1}} \left( u^{t-1} + \delta(\omega v^{t-1} + bu^{t-1}) \right)$$
$$= \delta \omega \tag{24}$$

$$\frac{\partial u^t}{\partial u^{t-1}} = \frac{\partial}{\partial u^{t-1}} \left( u^{t-1} + \delta(\omega v^{t-1} + bu^{t-1}) \right)$$
$$= 1 + \delta b \tag{25}$$

which results in:

$$\frac{\partial \mathbf{z}^t}{\partial \mathbf{z}^{t-1}} = \begin{bmatrix} 1 + \delta b & -\delta \omega \\ \delta \omega & 1 + \delta b \end{bmatrix} \tag{26}$$

If we now assume a sufficiently small $\delta$, we approximately obtain:

$$\frac{\partial \mathbf{z}^t}{\partial \mathbf{z}^{t-1}} \approx \begin{bmatrix} 1 & 0 \\ 0 & 1 \end{bmatrix} \tag{27}$$

We can see that the gradient state transition matrix (which is multiplied with the back flowing gradient at each time step) provides the characteristic of an identity matrix. This essentially means that the gradient is kept alive within the resonator during BPTT and stabilizes the gradient flow through the input gate. In practice, $\delta$ is usually not 0 and therefore the gradient slowly decays depending on $b$ and is rotated with $\omega$.

### A.3 RECURRENT SPARSITY

We studied the robustness against ablations of recurrent connections which we call "recurrent sparsity". This experiment can provide information about the learning progress of the model. The restricted access to memory gives indications on the self-sufficiency of the internal state. We compare the recurrent sparsity at 25%, 50%, 75%, and 100%, where the weights are disabled randomly with the respective percentage. The performance of the RG-RNN in Figure 5 largely retains a similar pattern as in the SMNIST experiment across all sparsity rates. A slight impact to convergence and performance can be observed starting at 75% sparsity with an even more pronounced impact when no recurrences are retained, essentially blocking the access to memory.

The LSTMs performance was influenced by all sparsity rates, as evident from Figure 6. First slightly decreasing the convergence and increasing the variance between runs at 25% and 50% sparsity. At 75% sparsity the convergence is improved with only a small impact in accuracy. Lastly, removing all recurrent connections has the greatest impact on the LSTM, severely reducing its accuracy. The performance of the RG-RNN and the LSTM can be taken from Tables A6 and A7, respectively.

The internal dynamics of the resonator seem to enable the RG-RNN to learn more effectively without the utilization of a memory. The hidden state sequence, which is masked internally, builds oscillations which give additional information to the RG-RNN when present. The same oscillations seem to be required for the LSTM to effectively learn the presented problem leading to a drop in performance when ablated. An interesting addition is the increased convergence of the LSTM at 75% recurrent sparsity. Unfortunately, a closer investigation was out of the scope of this paper.

### A.4 SUPPLEMENTARY FIGURES

The results of the MGA experiment from Section 4 can be taken from Figure 7. Additionally, the full loss and accuracy results for all experiments can be taken from Figures 8, 9, 10, and 11.

Table A6: Averaged results of the RG-RNN on SNMIST with sparse recurrencies. The models were trained with the corresponding percentage of masked recurrent connections.

| Metric | Sparsity 25% | Sparsity 50% | Sparsity 75% | Sparsity 100% |
|---|---|---|---|---|
| Train loss | 0.5977 | 0.6078 | 0.6325 | 0.9838 |
| Train accuracy | 99.80% | 99.72% | 99.35% | 84.55% |
| Test loss | 0.6549 | 0.656 | 0.6804 | 0.9863 |
| Test accuracy | 98.51% | 98.58% | 97.99% | 84.30% |

Table A7: Averaged results of the LSTM on SNMIST with sparse recurrencies. The models were trained with the corresponding percentage of masked recurrent connections.

| Metrics | Sparsity 25% | Sparsity 50% | Sparsity 75% | Sparsity 100% |
|---|---|---|---|---|
| Train loss | 0.6553 | 0.699 | 0.6612 | 1.848 |
| Train accuracy | 98.95% | 97.66% | 98.68% | 31.46% |
| Test loss | 0.7028 | 0.7165 | 0.6852 | 1.843 |
| Test accuracy | 97.28% | 96.89% | 97.83% | 32.02% |

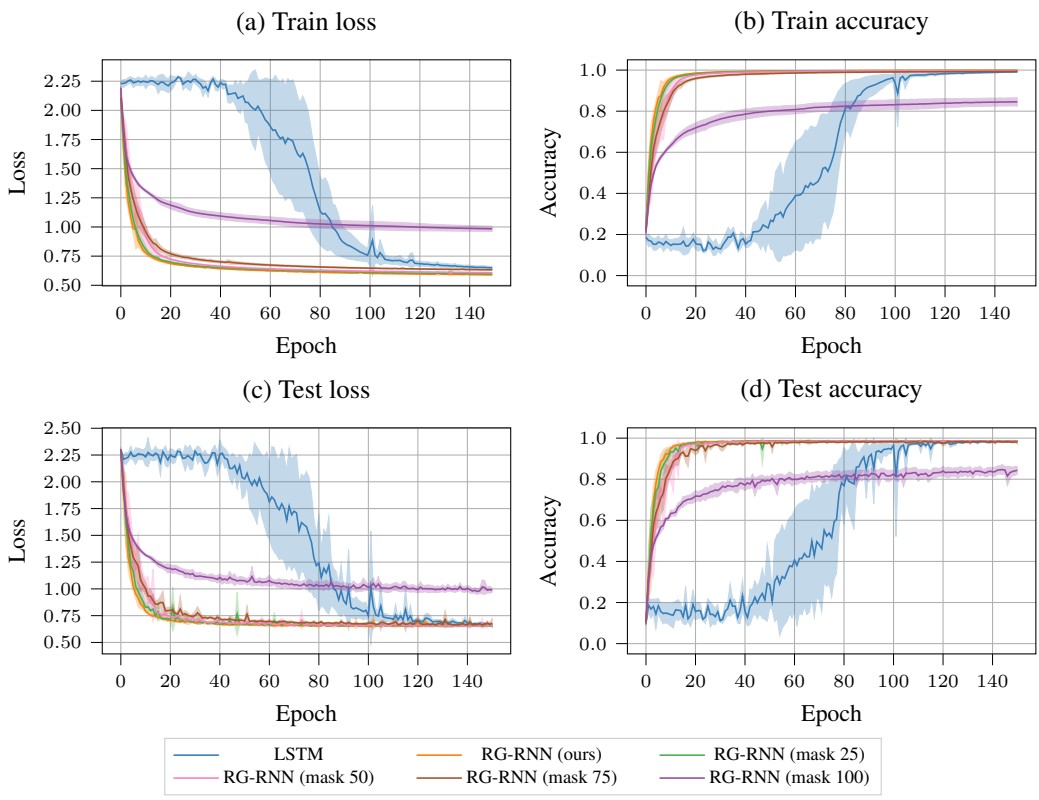

Figure 5: Experiment for **Sparse recurrencies** on **SMNIST** for the **RG-RNN**. Shown are averaged measures and standard deviation of the LSTM (blue) and the RG-RNN (orange) for (a) train loss, (b) train accuracy, (c) test loss, and (d) test accuracy.

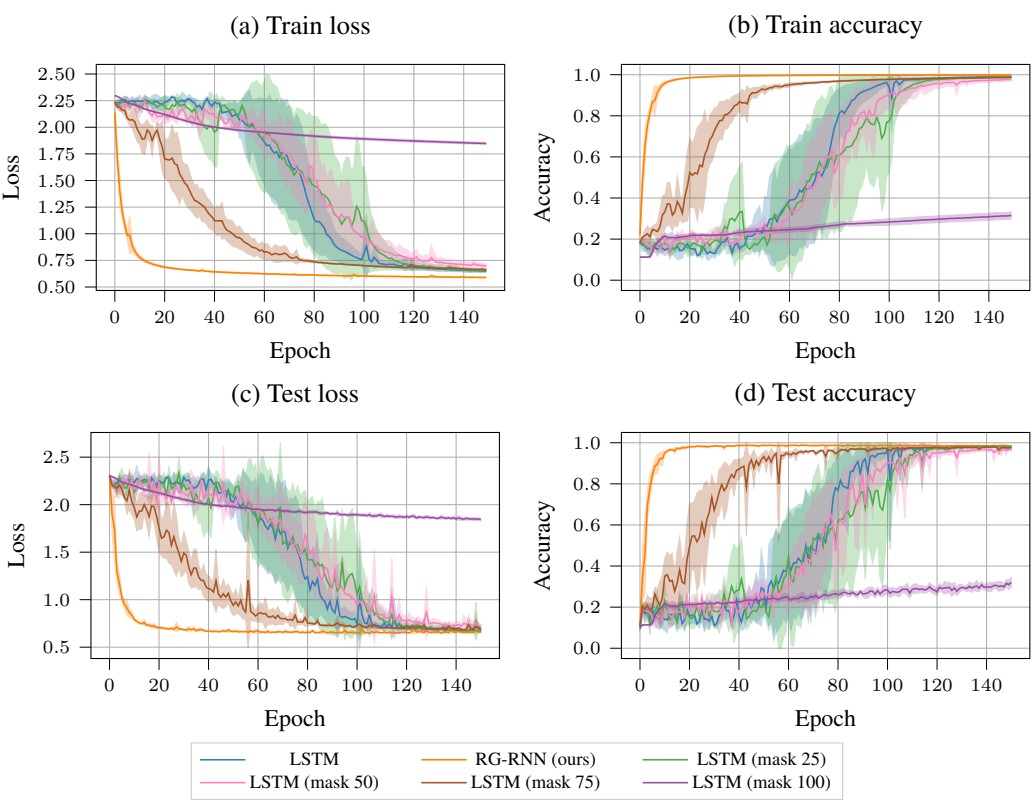

Figure 6: Experiment for **Sparse recurrencies** on **SMNIST** for the **LSTM**. Shown are averaged measures and standard deviation of the LSTM (blue) and the RG-RNN (orange) for (a) train loss, (b) train accuracy, (c) test loss, and (d) test accuracy.

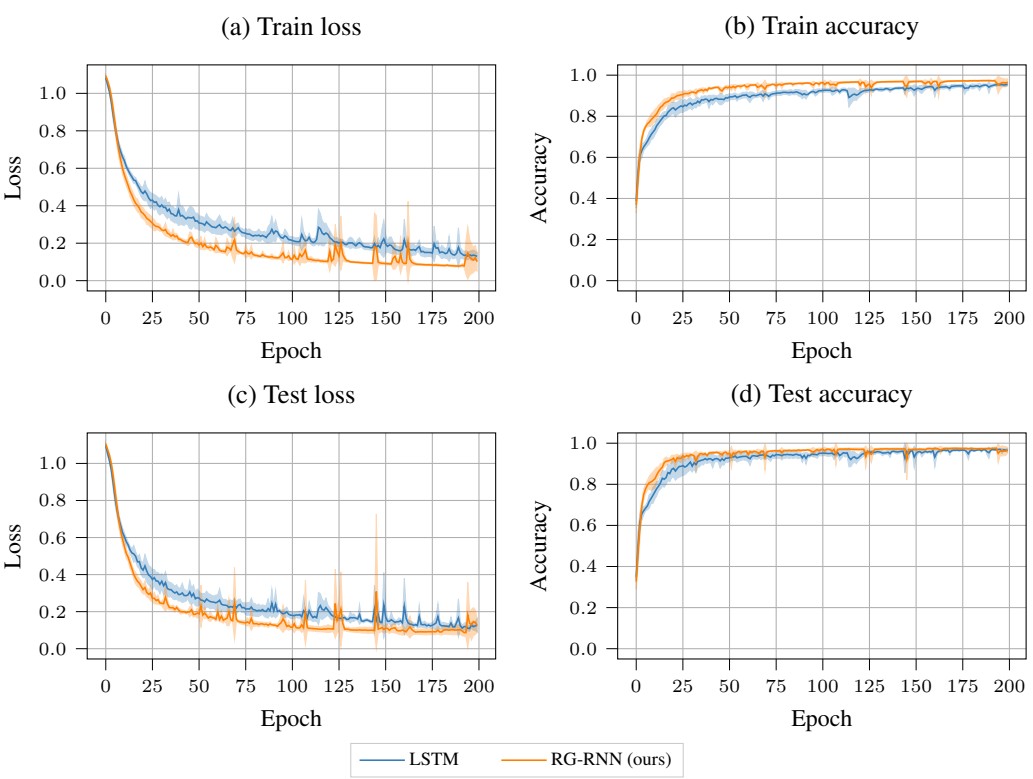

Figure 7: **MGA** experiment. Shown are averaged measures and standard deviation of the LSTM (blue) and the RG-RNN (orange) for (a) train loss, (b) train accuracy, (c) test loss, and (d) test accuracy.

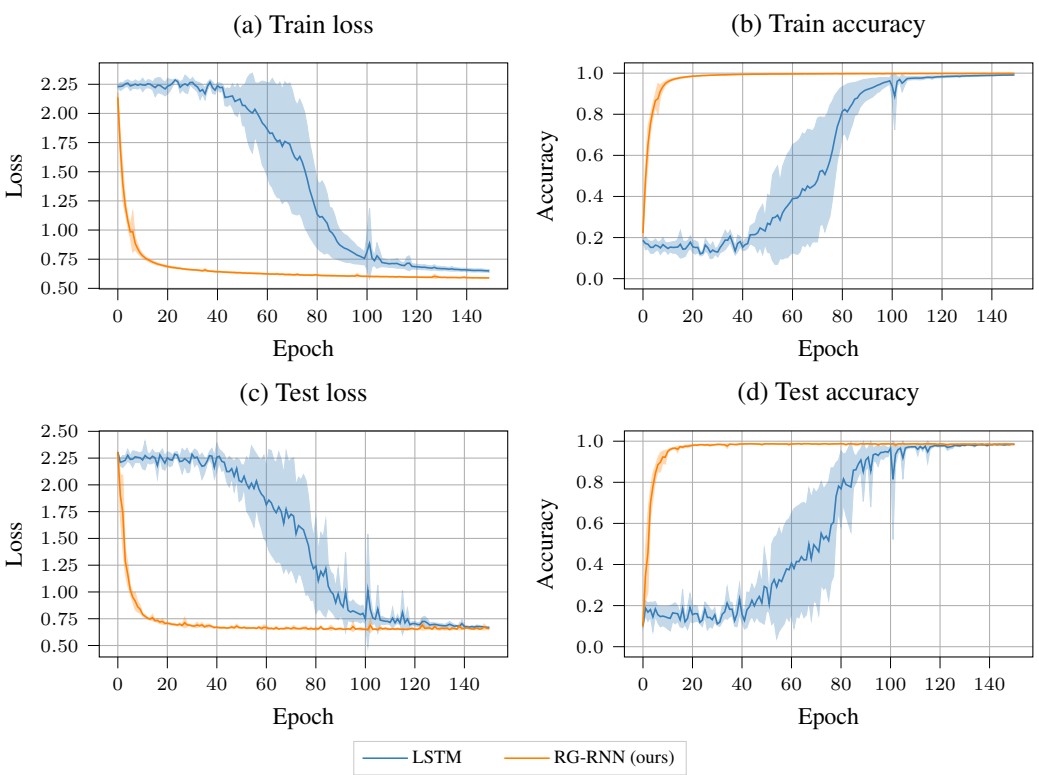

Figure 8: **SMNIST** experiment. Shown are averaged measures and standard deviation of the LSTM (blue) and the RG-RNN (orange) for (a) train loss, (b) train accuracy, (c) test loss, and (d) test accuracy.

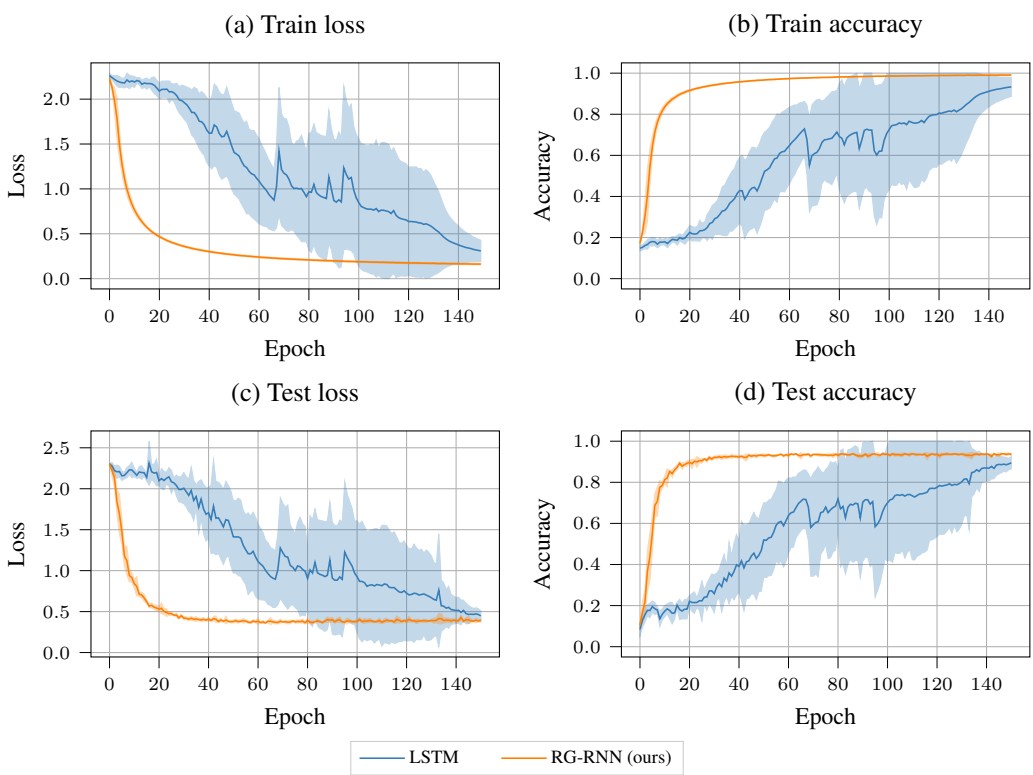

Figure 9: **Permuted MNIST** experiment. Shown are averaged measures and standard deviation of the LSTM (blue) and the RG-RNN (orange) for (a) train loss, (b) train accuracy, (c) test loss, and (d) test accuracy.

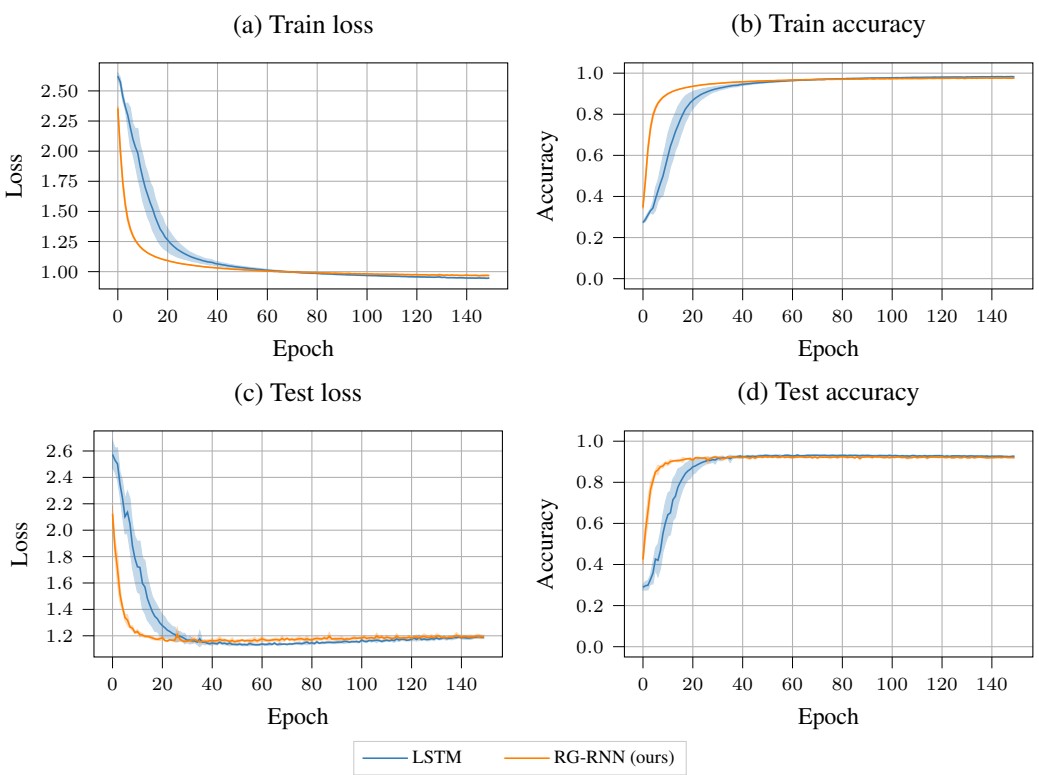

Figure 10: **Speech Commands V2** experiment. Shown are averaged measures and standard deviation of the LSTM (blue) and the RG-RNN (orange) for (a) train loss, (b) train accuracy, (c) test loss, and (d) test accuracy.

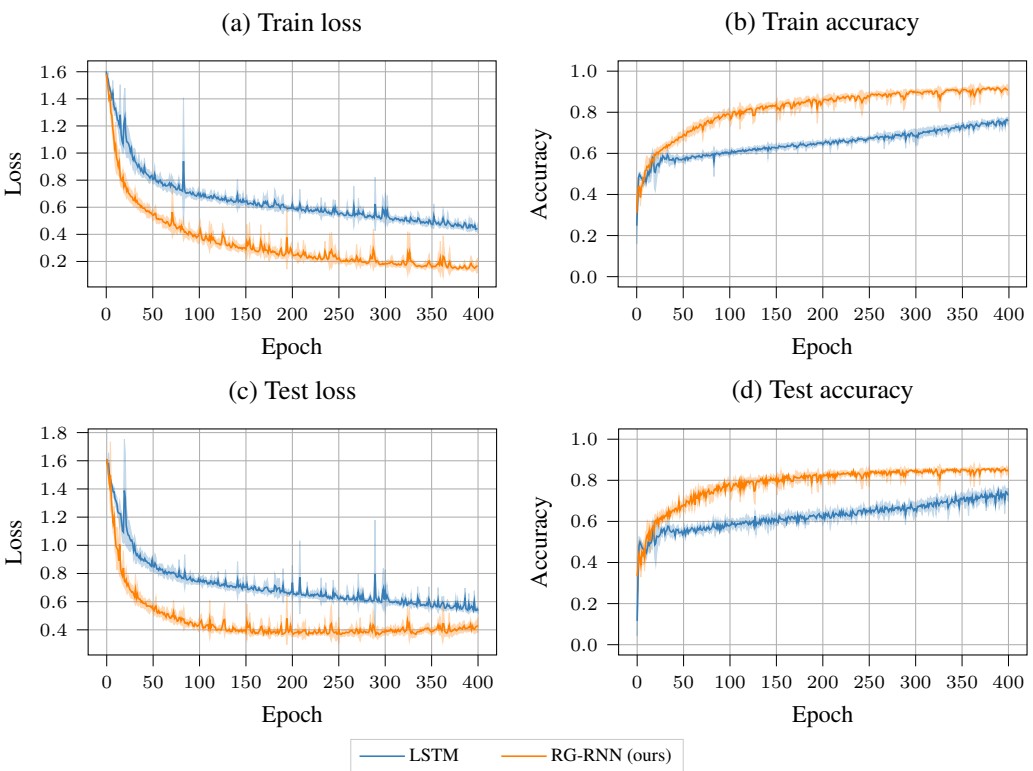

Figure 11: **QTDB** experiment. Shown are averaged measures and standard deviation of the LSTM (blue) and the RG-RNN (orange) for (a) train loss, (b) train accuracy, (c) test loss, and (d) test accuracy.

