# OpenReview forum: "Resonator-Gated RNNs"
_ICLR.cc/2024/Conference — Submitted to ICLR 2024_

### Official Review · Reviewer_KRtM · 2023-10-18

**Soundness:** 2 fair
**Presentation:** 4 excellent
**Contribution:** 2 fair
**Rating:** 3
**Confidence:** 4

**Summary:**

Gated RNNs fail to learn periodic functions efficiently. To address this limitation, the authors augment the LSTM architecture with a module leveraging the subthreshold dynamics of resonate-and-fire neurons. They show that this mechanism is sensitive to periodic inputs and that it improves the performance of the base architecture on a few datasets.

**Strengths:**

The introduction of the method is clear and using resonate-and-fire neurons to improve performance on periodic data is a cute idea.

**Weaknesses:**

Overall, the main weakness of the paper is that it does not really test its claims thoroughly. For example, one claim is that the improved performance is due to the ability to capture periodic dependencies. From the experiments described, it is impossible to understand if this is true or if the improved performance comes from additional parameters.

**Questions:**

How does your work relate to recent literature on linear diagonal state-space models (for example Orvieto et al 2023, Resurrecting recurrent neural networks for long sequences, and references mentioned in the paper)? To me, those results suggest that resonate-and-fire type of mechanisms greatly simplify gradient-based learning in general, and not the learning of periodic functions specifically.

---

> ### Author Response · Authors · 2023-11-22
> **Reviewer Answer**
>
> Thanks to reviewer for the important comments.
>
>  Addressing the Questions:
>
> We see that we did not sufficiently embed our method into the current landscape especially with respect to the advances that were published in the last years. We addressed this in the manuscript and added respective references.
>
> Addressing the Weaknesses:
>
> We see that we currently do not stress how the number of parameters is influenced by the addition of the resonator. We added a paragraph that is aimed to clarify this. Here we want to emphasize that the number of parameters of a RG-LSTM with 128 units is significantly smaller that an LSTM with 129 units. Because the addition of a single hidden neuron is typically negligible the change in behavior must lie in the added inductive bias.
>
> The benefits seem to arise from an interplay of a 'periodic gating' that feed (and receives its gradient from) a stable 'rotating' memory. We added a derivation that essentially shows that the gradient flow within the resonator is stably maintained - it thus acts like an additional error carousel (cf. CEC in LSTMs). While this does not prove the extraction or learning of frequencies it gives some indication that this can be the case as the inductive bias of the resonator primes it towards periodically picking up information and adding it to the memory.

---

### Official Review · Reviewer_ZNzp · 2023-10-30

**Soundness:** 4 excellent
**Presentation:** 2 fair
**Contribution:** 2 fair
**Rating:** 6
**Confidence:** 3

**Summary:**

The paper "RESONATOR-GATED RNNS" proposes to add two discretized resonating differential equations to the LSTM gate equations. The resulting RG-LSTM is compared to an LSTM baseline on the sequential MNIST,
permuted sequential MNIST, speech Commands VS and the OTDB dataset of physiological signal recordings.

**Strengths:**

- Cell convergence results appear to be competitive.
- The experimental evaluation features mean values and standard deviations over multiple runs.

**Weaknesses:**

- Recurrent networks used to be the go to choice for sequence modelling. In the deep learning book (https://www.deeplearningbook.org/), chater 10 bears the title "10 Sequence Modeling: Recurrent and Recursive Nets" Consequently, the paper calls RNNs the go to standard for sequence modelling. However a lot has happend since 2016. Attention based systems like transformers have since emerged as a popular alternative to RNNs for sequence modelling tasks, unfortunately this development is not discussed in the related work. In the vision domain Trainsformers are known to struggle on small data-sets (https://proceedings.neurips.cc/paper/2021/file/c81e155d85dae5430a8cee6f2242e82c-Paper.pdf). Perhaps the same is true for sequential data? This could be possible way to extend the related work without having to run additional experiments.

**Questions:**

- What does the OTDB acronym mean?
- Which seed values have been used? Without the seeds, it won't be possible to reproduce the paper's experimental results exactly.
- What are the trainable parameters for the LSTM and RG-LSTM cells for each experiment?
  Was the cell state size identical in both cases? Does this mean that the RG-LSTM cell has more trainable parameters?
- Some authors suspect extra weights improve convergence (https://arxiv.org/abs/1803.03635 ).
- If the number of trainable weights was not the same, is the comparison fair?

---

> ### Author Response · Authors · 2023-11-22
> **Reviewer Answer**
>
> We thank the reviewer for the comments and questions.
>
>
> Please apologize that we did not further elaborate this acronym. QTDB stands for QT Database where QT is the waveform interval in ECGs. We added a remark in the manuscript.
>
> We will provide respective seeds with publication of the code. But we do not agree with this argument: if the method is sufficiently stable, randomized trials (which can be replicated) that deliver reasonable performance are more convincing than choosing some well working seeds. Moreover, in contrast to the LSTM, the standard deviation for all random runs of the RG-RNN is extremely narrow.
>
> We agree that - even if unlikely - a few additional weights could in principle make a difference. While the number of parameters was higher for the RG-LSTM the difference is less pronounced than comparing the LSTM with an LSTM that has just a single additional hidden unit. To clarify we will report the number of parameters for LSTM and RG-LSTM. In addition we can retrain an LSTM with an additional hidden unit e.g. 129 for SMNIST. The performance impact of this will be negligible based on first evaluations.
>
> Addressing the Weaknesses:
>
> We further extend on current sequence learning advances and especially on the connections to attention-based models and recent SSMs and LRU.

---

### Official Review · Reviewer_kRNM · 2023-10-31

**Soundness:** 2 fair
**Presentation:** 3 good
**Contribution:** 2 fair
**Rating:** 5
**Confidence:** 4

**Summary:**

This article addresses the challenge of RNNs in handling periodic data. To overcome this, the authors introduce the resonate-and-fire neuron and propose the Resonator-Gated RNN (RG-RNN), which outperforms LSTM on multiple periodic datasets.

**Strengths:**

1. The article is well-written and easy to understand.
2. The method proposed is simple yet effective.

**Weaknesses:**

My main concern about this article is whether the comparison is comprehensive enough. For example, there have been new types of RNNs such as S4, S5, and LRU, so it would be interesting to compare the performance of these methods. Additionally, introducing the Transformer as a baseline can also help readers better understand the strengths and weaknesses of each model on periodic datasets.

**Questions:**

Add S4, S5, LRU and transformer as baseline.

---

> ### Author Response · Authors · 2023-11-22
> **Reviewer Answer**
>
> We thank the reviewer for the critical reflection of our work.
>
> We particularly agree that including recurrent long-range sequence models could be insightful. We would be happy to include respective results in the manuscript. However, we think that it does not really make sense to incorporate transformers. These two architectures are fundamentally different, whereas our research question was to study, what a resonating circuit adds to an RNN, or more specifically, to an LSTM (the latter is still frequently used as a sequence learning model). In fact we demonstrate an immense gain of convergence speed and stability just with an moderate architectural modification without the need for additional explicit regularization.

---

### Official Review · Reviewer_UaUD · 2023-10-31

**Soundness:** 3 good
**Presentation:** 3 good
**Contribution:** 3 good
**Rating:** 6
**Confidence:** 3

**Summary:**

The paper proposes to modify the LSTM architecture with a `resonator-gate` with the aim of improving the detection of periodic components of an input sequence. The resonator is defined based on the discrete time version of a `resonate-and-fire` neuron model and is of similar computational cost as the existing LSTM gates. The authors evaluate the performance of RG-RNN against LSTMs on four time-series dataset and demonstrate greatly improved training stability, convergence behavior and performance in almost all cases.

**Strengths:**

- Improving existing architectures for sequence prediction is important problem. As the authors note, LSTMs and GRUs have been the de facto standards for quite some time and an improvement in training performance will be greatly benificial.
- The paper is overall well written and easy to follow.
- Idea is simple, intuitive and easy to implement and test.
- Experimental results on the 4 datasets are impressive, showing much-faster convergence and stability  in training (Figure 3) and improved performance (Table 1).

**Weaknesses:**

While experimental section overall is sound and (seemingly) reproducible, it can be greatly improved with experiments on time-series data from other data domains (ex. sensor data, from UCI Data-repo). Given the simplicity of the proposed method (both conceptual and in implementation) and the popularity of LSTM/GRU cells, an larger evaluation demonstrating stability/convergence behavior will be greatly strengthen the paper.

**Questions:**

- While Figure~1 demonstrates how the discrete time resonator dynamics behaves, it would be interesting to see this also as part of the LSTM architecture. For example, working with a  1D synthetic sequence prediction task (say on some periodic signal), is the difference in performance between an LSTM and RG-LSTM evident?
- Can the modified gating mechanism be applied to GRU as well? How does the performance compare?
-  'prioritize retaining static relevant information': Could you elaborate on `static relevant information`?

---

> ### Author Response · Authors · 2023-11-22
> **Reviewer Answer**
>
> We thank the reviewer for the value comments and suggestions. Please find our responses below.
>
> To properly answer the reviewers first question we would need some more information of the type of experiment that was in mind. It would be possible to perform the experiment conducted for Figure 1, but it would be difficult to make any statements about the output because it is randomly initialized and untrained. It would in fact be interesting to evaluate the behavior of the RG-RNN in regression tasks (such as time-series forecasting), which is another line of research for us. Our intuition is, however, that the resonator functions primarily as a detector and does not necessarily facilitate the creation of oscillations.
>
> The resonator-based gating mechanism can be applied to GRUs as well. We think that it would best be applied to the update gate since it regulates how the new inputs are combined with the current hidden state. This is definitely an interesting question and something we will look into - we made a respective comment in the paper. We would assume a similar effect to the network's convergence. But as there is no fundamental and consistent advantage of the GRU over LSTM (there is actually a tendency of the latter working slightly better), we expect the same results when comparing RG-GRUs and RG-LSTMs.
>
> We see that the term 'static relevant information' is not accurate enough. The type of information that is retained by RNNs is more clearly described as ``transient'', it arises and/or disappears during the sequence.
>
> Addressing the weaknesses:
>
> We see the demand for more results on another dataset and hence added an analysis of gait trials from sensor data provided on UCI. We agree that adding another data domain can emphasise the strengths as well as give clearer indication of weaknesses of the method.

---

### Author Response · Authors · 2023-11-22
**Modifications to the Manuscript**

We would like to thank all the reviewers for their valuable comments, suggestions, and interesting thoughts based on which we made several changes and additions to improve our paper. These are in particular:

- We included a new dataset (MGA) in our experiments.
- We added a derivation of the recurrent gradient flow within the resonator to give an indication of its convergence. It can be found in the appendix
- We added an overview of the trainable parameters for every model and performed an exemplary experiment which showcases the performance of the RG-LSTM against an LSTM with more trainable parameters. Both can be found in the Appendix
- We added more references to related work including SSMs and LRU.
- We also did minor clarifications in the text.

---

### Meta-Review · Area_Chair_Ah5S · 2023-12-12

**Metareview:**

This paper proposes an architectural variation on the familiar LSTM and GRU memory units called a resonator-gated RNN. The method outperforms vanilla LSTMs on several tasks. The reviewers note that while LSTM and GRUs are standard "memory cells" in the RNN literature, the broader literature on sequence modeling has moved in a very different direction since 2017 with the advent of transformers and note that the paper lacks any discussion of Transformers or comparison to Transformer-based baselines. The authors say "does not really make sense to incorporate transformers" but this doesn't quite add up. If the claimed benefit is in terms of empirical performance on tasks of interest, it's worth asking whether there's any reason to believe the method would actually be useful. The reviewers also asked for a more thorough empirical evaluations with more experiments on a broader set of datasets. Overall, based on the scores the paper is in reject territory. The authors wrote somewhat anemic rebuttals and received relatively little engagement from reviewers. I take some responsibility for the lack of a more thorough discussion but also do not believe that the authors provided too substantial a reply to license a back and forth. In short, while I am comfortable recommending rejection, it seems like the work holds potential and the primary complaints of the reviewers can be addressed with a much more thorough set of baselines and experiments, as well as an elaborated exposition that takes the time to situate this work relative to the dominant approaches to sequence modeling in the current era.

**Justification For Why Not Higher Score:**

Missing baselines, insufficient experiments, no consideration of transformers.

**Justification For Why Not Lower Score:**

N/A

---

### Decision · Program_Chairs · 2024-01-16

Reject